# Fast and Purification-Free Characterization of Bio-Nanoparticles in Biological Media by Electrical Asymmetrical Flow Field-Flow Fractionation Hyphenated with Multi-Angle Light Scattering and Nanoparticle Tracking Analysis Detection

**DOI:** 10.3390/molecules25204703

**Published:** 2020-10-14

**Authors:** Roland Drexel, Agnieszka Siupa, Pauline Carnell-Morris, Michele Carboni, Jo Sullivan, Florian Meier

**Affiliations:** 1Postnova Analytics GmbH, Max-Planck-Straße 14, 86899 Landsberg, Germany; roland.drexel@postnova.com; 2Malvern Panalytical Ltd., Grovewood Road, Malvern, Worcestershire WR14 1XZ, UK; pauline.carnell-morris@malvern.com (P.C.-M.); michele.carboni@malvern.com (M.C.); jo.sullivan@malvern.com (J.S.)

**Keywords:** liposomes, exosomes, rabbit serum, cell culture medium, protein corona, electrical asymmetrical flow field-flow fractionation, nanoparticle tracking analysis, size separation, absolute number concentration, zeta potential

## Abstract

Accurate physico-chemical characterization of exosomes and liposomes in biological media is challenging due to the inherent complexity of the sample matrix. An appropriate purification step can significantly reduce matrix interferences, and thus facilitate analysis of such demanding samples. Electrical Asymmetrical Flow Field-Flow Fractionation (EAF4) provides online sample purification while simultaneously enabling access to size and Zeta potential of sample constituents in the size range of approx. 1–1000 nm. Hyphenation of EAF4 with Multi-Angle Light Scattering (MALS) and Nanoparticle Tracking Analysis (NTA) detection adds high resolution size and number concentration information turning this setup into a powerful analytical platform for the comprehensive physico-chemical characterization of such challenging samples. We here present EAF4-MALS hyphenated with NTA for the analysis of liposomes and exosomes in complex, biological media. Coupling of the two systems was realized using a flow splitter to deliver the sample at an appropriate flow speed for the NTA measurement. After a proof-of-concept study using polystyrene nanoparticles, the combined setup was successfully applied to analyze liposomes and exosomes spiked into cell culture medium and rabbit serum, respectively. Obtained results highlight the benefits of the EAF4-MALS-NTA platform to study the behavior of these promising drug delivery vesicles under in vivo like conditions.

## 1. Introduction

One of the key stages in therapeutics’ development is formulation optimization to select the right candidate, where analytical techniques support lead candidate identification for later evaluation in more complex formulations. Traditional methods for characterization of bio-particles and liposomal drug delivery carriers such as Dynamic Light Scattering (DLS) are widely accepted by regulators in pre-clinical formulation development [1,2,3,4]. The effort, time and cost associated with the selection of the right candidate for clinical trials are closely monitored by company investors. Challenges often occur when the carrier behavior in in vivo testing is significantly different from its previous in vitro activity. Therefore, characterization of bio-particles and drug carrier systems in conditions mimicking the in vivo environment is gaining an interest. In addition, easily and quickly obtaining an earlier understanding of their toxicological effects is attractive within the therapeutic development pipeline. The system complexity of the solutions necessary for testing remains the biggest challenge in identification and monitoring of the target particles in in vivo like conditions.

Similarly, when characterizing bio-particles extracted from various body fluids, e.g., exosomes or viruses, additional isolation and purification steps are necessary, which also require additional validation steps. Usually small in size, 30–150 nm [5,6], exosomes carry important messages coded in RNA or proteins which play a vital role in intercellular communication. Thanks to their ability to protect the cell cargo in body fluid, deliver the cargo to remote cells or back to the cell of origin and possibility of transporting infectious agents, exosomes are emerging as top biomarkers for a disease diagnostic (a so called “liquid biopsy”) [7,8,9,10]. Also, being comprised of a lipid bilayer and having an abundance of proteins on their surface that allows for easy uptake by cells, exosomes are being investigated as promising drug delivery materials [11,12].

While supporting biological assays, analytical techniques provide information on a sample under test specific conditions. In many cases for biological nanoparticles, the samples are a complex mixture, so isolation and purification processes are required to enable best measurement conditions for the technology, to allow high quality decision-making data to be generated [13,14,15]. Multi-technology characterization, meaning the combination of complementary analytical and bioassay techniques, aids in expanding knowledge of sample performance, however analysis and user time is valuable and plays an important role when working to tight deadlines. Automation and method hyphenation offer the potential to perform analyses and collect data quickly and effectively. The hyphenation of Electrical Asymmetrical Flow Field-Flow Fractionation (EAF4), Multi-Angle Light Scattering (MALS) and Nanoparticle Tracking Analysis (NTA) therefore is an attractive offer of multi-parameter characterization in a continuous analysis of very complex samples.

Field-Flow Fractionation (FFF) comprises a family of elution-based separation techniques capable of a rapid and high efficiency separation of suspended and dissolved samples in the size range of 1 nm to several micrometers [16,17,18]. One sub-technique of FFF is EAF4, which is a variant of the most common FFF-technique: Asymmetrical Flow Field-Flow Fractionation (AF4). In EAF4, separation is achieved in a narrow, ribbon-like channel with a semipermeable ultrafiltration membrane (accumulation wall) at the channel bottom. Inside this channel, a laminar flow is applied exhibiting a parabolic flow profile. Perpendicular to the laminar channel flow, a second flow (cross flow) and an electric field are applied simultaneously thereby inducing separation of sample constituents by size and charge (Appendix A). Besides offering high-resolution separation and access to particle size distribution and surface charge (Zeta potential) of a respective sample, the possibility to purify (i.e., to remove matrix components smaller than the molecular weight cutoff of the ultrafiltration membrane via the cross flow) and separate matrix components from the analyte of interest renders EAF4 (just like AF4) a promising tool for an online purification of complex biological samples prior to further multi-detector analysis [19,20].

NanoSight NTA uses a laser light source to illuminate particles in a liquid suspension, moving under Brownian motion. The light scattered by particles is then collected through specially configured optics and particle movements are registered with a high sensitivity sCMOS camera. NTA software tracks the Brownian motion of particles in order to calculate mean square displacement for hydrodynamic diameter calculation based on the Stokes-Einstein equation. All particles are measured individually and simultaneously for high resolution size and concentration data within an applicable size range of 10 nm–1000 nm depending on their specific optical properties. NanoSight NTA can be used with appropriate flow conditions for improved sampling without compromising data quality [21,22].

Over the past years, both FFF and NTA have significantly grown in popularity for the analysis of complex biological samples including extracellular vesicles such as exosomes [23,24,25,26,27,28,29], and liposomes [4,30,31,32,33]. It was therefore just a matter of time until both technologies were coupled in order to combine the advantages of both worlds. In 2015, Bartczak et al. presented for the first time the potential of a hyphenation of AF4 and NTA for the analysis of silica nanoparticles using a start-stop mode in order to circumvent the challenges associated with the vastly divergent flow rates of AF4 and NTA [34]. Recently, in a proof-of-concept study, the first true on-line coupling of AF4 and NTA was realized by Adkins et al. using a splitter manifold connection. This setup was successfully applied for the characterization of polystyrene particles, gold nanorods and hexagonal boron nitride nanosheets in simple aqueous matrices [35].

In this paper, we describe the online hyphenation of EAF4 and NTA offering the comprehensive physico-chemical characterization of highly heterogeneous samples in biologically relevant media containing liposomes and exosomes. Liposomes hereby served as model system for exosomes, which are considered a low refractive index sample that is usually challenging to analyze by NTA. The separation of the particles of interest, and detailed sample characterization is possible by effective reduction of the flow from the EAF4 system using the slot-outlet option [36,37] and a flow splitter to deliver continuous fractions for NTA analysis. EAF4 enables separation of the particles of interest from the constituents of a complex solution, offering an alternative to traditional (offline) purification techniques, while analysis with EAF4 detectors including Multi-Angle Light Scattering (MALS), followed by NTA confirms the separation and provides detailed information about the particle size, Zeta potential and number concentration of a respective sample.

## 2. Results and Discussion

### 2.1. Offline Hyphenation of EAF4-MALS and NTA

In a first step, the hyphenation of EAF4-MALS and NTA was validated against well-characterized polystyrene nanoparticles, which served as a model particle system to evaluate the suitability of this novel setup for the analysis of nanoparticles in complex biological media. Therefore, two different samples of 100 nm polystyrene particles (PS100) were separated by EAF4 using fractionation method A (Appendix A) and a fraction containing the peak maximum of the PS100 sample was collected for every measurement (Appendix A). The first sample contained PS100 in ultrapure water (UPW) and the second sample contained the same concentration of PS100, but it was diluted with Dulbecco’s Modified Eagle’s Medium (DMEM) that also included 10% FCS (fetal calf serum). The challenge of analyzing samples containing cell culture media components by NTA in off-line light scatter mode is demonstrated in Appendix A. The picture illustrates the detection of a high concentration of culture media components that complicates the differentiation of scattered light from matrix constituents and the nanoparticles of interest, which can lead to error-prone results.

The EAF4-MALS fractograms of PS100 in UPW and DMEM are displayed in Figure 1. Fractionation was achieved with an excellent sample recovery of 87.2% ± 1.4% and 100.1% ± 0.9%, respectively. The retention time of the peak maximum is shifted to higher retention times by a few minutes when the particles are suspended in DMEM, while exhibiting a comparable peak shape. The radius of gyration (R_g_) distribution was shifted to higher retention times as well, but no changes in the size range or the R_g_ distribution was observable, since MALS signals are usually insensitive to small surface changes induced by the creation of a protein corona onto a dense particle core. Nonetheless, both results indicate an increased hydrodynamic diameter (D_h_) for the sample containing the cell culture medium most likely due to the formation of a protein corona on the surface of the PS particles [38].

In a subsequent analysis with NTA, the particle number concentration and D_h_ were determined using Script 0 (see Appendix A for NTA setting details). The details of the parameters used to perform the offline NTA measurements are listed in the Materials and Method section and in the Appendix A. Two different concentrations of each sample were injected in duplicate into the EAF4-MALS system to assess the best conditions for separation and the optimum concentration range for the NTA measurements. Each fraction was then analyzed in triplicate by NTA. The results are listed in Table 1.

The particle number concentration increased proportionally with the injected amount of sample, with a standard error below 5%. Secondly, the NTA measurements showed repeatable results for the D_h_ with very low standard errors for both injected concentrations.

The second sample, which contained a complex matrix of DMEM cell culture medium showed comparable particle number concentrations, but D_h_ increased significantly. This can most likely be explained by the presence of various proteins, which adhere to the particles and form a protein corona on the PS beads’ surface [38]. These results also confirm that the shift in EAF4 retention time outlined above is indeed related to an increase in D_h_ rather than being an artefact from DMEM constituents significantly changing the surface properties of the used EAF4 membrane.

The DMEM cell culture medium is a highly complex matrix that strongly scatters laser light, which can interfere with the scattered light from the particles [22,26]. A full separation of the PS100 beads from proteins in the sample by a precedent EAF4 purification step is a viable way to reduce the interfering protein background leading to individual PS100 particle fractions that are much better amenable to NTA analysis.

### 2.2. Online Hyphenation of EAF4-MALS and NTA

After highlighting the potential of the combination of EAF4-MALS and NTA in an offline setup, the online coupling of EAF4-MALS and NTA was tested and validated on previously investigated samples. The optimized fractionation conditions are summarized in Appendix A. The separation of the PS100 sample was carried out using “Fractionation method A” (see Appendix A). 

Figure 2 displays the fractograms overlaying the obtained particle number concentration with the determined D_h_. The particle number concentration for each measurement point over the complete peak was in the optimum concentration range for the NTA measurement, which lies in-between 1 × 10^6^ particles mL^−1^ and 1 × 10^9^ particles mL^−1^ [39]. The peak maximum of the particle number concentration is slightly shifted to higher retention times in the same way as it was shown for the MALS 92° signal in Figure 1.

An average D_h_ of 113.3 nm was obtained for the PS100-DMEM sample, and 96.7 nm for the PS100 sample, respectively. This is in excellent agreement with the results obtained from the offline measurements again strongly indicating an increase in size due to the formation of a protein corona on the PS100 beads surface. The above described observations were also investigated by calculating the Zeta potential, which is a function of the surface charge, from observed EAF4 retention time shifts. For the PS100 sample, a Zeta potential of −48.0 mV was determined, whereas a Zeta potential of −37.1 mV for the PS100-DMEM sample was obtained thus again indicating changes in the surface composition of the PS100 beads. Compared to the MALS signal, the peak width of the NTA-signal was slightly increased, which can be explained by band broadening effects caused by the geometry of the NanoSight Low Volume Flow Cell (LVFC) and the additional tubing, although the LVFC inlet was connected to the “short” end, which is illustrated in Appendix A. Summarizing the described results, the hyphenation of EAF4-MALS and NTA provides helpful additional information on the characterization of PS beads in highly complex matrices that are challenging to assess using NTA as a standalone technique. The obtained results are displayed in Table 2 also highlighting the very good recovery rate of above 90% for the fractionation that was achieved for both samples.

In a third step, a liposomal drug carrier (liposomal Doxorubicin HCl), used as a model low refractive index sample, was characterized using the hyphenated EAF4-MALS-NTA setup applying “Fractionation method B” (Appendix A). Figure 3 shows the obtained EAF4-MALS fractograms including the calculated R_g_ distribution ranging from around 24 nm up to 41 nm. Under the optimized fractionation conditions, a high recovery rate of more than 87% was achieved for both samples. All results of the respective EAF4-MALS-NTA experiments are summarized in Table 3.

To obtain meaningful NTA results, the concentration of the liposomal Doxorubicin HCl had to be decreased to around 0.67 mg L^−1^ by dilution with the carrier solution prior to the EAF4 injection. Two samples were prepared; one contained the liposomes diluted in the carrier solution, whereas the second sample was diluted in DMEM cell culture medium. The obtained fractograms displaying the particle number concentration and D_h_ are shown below in Figure 4. At the peak maximum, a particle concentration of around 7.50 × 10^8^ particles mL^−1^ was obtained for the high concentration sample with a D_h_ ranging from around 65 nm to 95 nm across the whole peak. It was observed that the presence of proteins in the DMEM cell culture medium did not significantly affect the D_h_ of the liposomes indicating no measurable formation of a protein corona on the liposomes’ surface at least for the duration of the whole experiment (<2 h). These findings are also in agreement with data published by Gioria et al. and Hu et al. [40,41]. However, the observed changes in Zeta potentials of liposomal Doxorubicin HCl diluted either in 0.5 mM sodium chloride solution (−34.6 mV ± 1.5 mV) or DMEM cell culture medium (−45.2 mV ± 1.5 mV) clearly indicate changes in the surface composition of the respective sample. Instead of proteins, this observation might in this case be related to the adsorption of smaller DMEM components such as e.g., amino acids or other electrolytes, that don’t significantly contribute to an increase of the D_h_ of the liposomes.

Additionally, from the relation of the endpoints of both sizing techniques, MALS and NTA, a statement about the particle shape can be made. At the intensity peak maximum (MALS 92° signal) a radius of gyration R_g_ of 30.4 nm and at the concentration maximum (NTA) a hydrodynamic diameter D_h_ of around 76.9 nm was calculated yielding a ratio of D_g_/D_h_ of 0.79 with D_g_ = diameter of gyration (i.e., 2 × R_g_). This indicates a filled, respectively solid, spherical morphology, which is also confirmed by the MALS light scattering data (Appendix A) [42,43].

Due to its increasing importance in nanomedical and diagnostic applications, an exosome sample was also investigated. Here, an exosome standard, which was extracted from human urine, was prepared both in phosphate buffered saline (PBS) buffer and in rabbit serum. Just like the DMEM medium, the rabbit serum represents a highly complex mixture containing a variety of proteins and electrolytes. Hence, in order to exclude matrix-induced interferences, a comprehensive EAF4-MALS-NTA investigation was performed instead of a single NTA measurement using the separation conditions described in Appendix A (“Fractionation method C”). Unfortunately, a determination of the recovery rate was not possible in this case due to the very low UV-vis-signal intensity of the exosomes, however, sufficient recovery rates can be assumed based on the monitored MALS and NTA signals. In Figure 5a, fractograms with an overlay of D_h_ and the particle number concentration of the exosome standard are displayed. The exosome D_h_ distribution ranged from around 43 nm up to a maximum of 150 nm, showing higher polydispersity than both the PS100 beads and the liposomal Doxorubicin HCl sample. The NTA results also confirmed the excellent repeatability of the EAF4 separation with around 7.4 × 10^8^ particles mL^−1^ for the highest injected concentration. In addition, the distribution of R_g_ increased from around 23 nm to 100 nm (Figure 6a) while the hydrodynamic diameter D_h_ in the particle concentration signal maximum (NTA) was determined to 98.2 nm. Compared to a R_g_ of 37.9 nm (i.e., D_g_ = 75.8 nm) at the MALS 92° signal maximum, a ratio of D_g_/D_h_ of roughly 0.77 was calculated representing a filled sphere, which is in line with the MALS scattering data (Appendix A).

In Figure 6b the R_g_ distribution across the 92° MALS signal peak in serum ranging from around 23 nm to 100 nm is illustrated. Obtained MALS data confirm the successful separation of rabbit serum from exosomes. Furthermore, the R_g_ distribution of the exosomes was not affected after spiking to rabbit serum. On the other hand, a D_h_ distribution from roughly 35 nm up to 90 nm with increasing D_h_ variations in serum was observed. The particle concentration maximum of the exosomes in rabbit serum compared to the exosome standard measurement in buffer shifted slightly to smaller retention times, in contrast to the local maximum of the 92° MALS signal between 30–50 min, which showed a slight shift to higher retention times (Figure 6b). After initial measurements of the serum matrix (blank) (see also Figure 6b) a meaningful size fraction could be determined that ranged in the lower size range of the exosomes. As a consequence the determined exosome concentration in serum increased significantly compared to the expected values and the particle concentration maximum as well as the D_h_ in the particle concentration maximum (Figure 5a,b) shifted towards lower values due to the weighting by the particles present in the serum matrix (see Table 4). The obtained results are summarized in Table 4.

The application of an electrical field during EAF4 separation led to a substantial drop in the recovery rate of both exosome samples while the respective field-off peaks in the EAF4-MALS fractograms increased considerably indicating either increased particle-membrane interactions, sample agglomeration or even aggregation. Unfortunately, the lack of exosome stability under these conditions rendered the determination of the Zeta potential meaningless and therefore, no respective values are displayed in Table 4 (n.d. not determinable). A deeper investigation of the potential reasons for these observations was beyond the scope of this publication and will be subject of further studies.

## 3. Materials and Methods

### 3.1. Chemicals and Samples

UPW was obtained from a Milli-Q system (Integral 5 system, Merck KGaA, Darmstadt, Germany) and filtered with a vacuum filtration unit through a 0.1 µm pore membrane (Durapore, Merck Millipore Ltd., Tullagreen, Ireland). For optimum fractionation and high recoveries, the carrier solution was adjusted to the specific sample type and fractionation problem.

Sodium chloride and sodium azide were purchased from Avantor Performance Materials Poland S.A. A PBS buffer solution at pH = 7.40 was prepared consisting of 10 mM phosphate salts (potassium dihydrogen phosphate and sodium hydrogen phosphate salts, obtained from Th. Geyer GmbH Co. KG, Renningen, Germany, respectively C. Roth GmbH Co. KG, Karlsruhe, Germany), 2.7 mM potassium chloride (C. Roth GmbH Co. KG, Karlsruhe, Germany), 137 mM sodium chloride and 0.02 wt% sodium azide. The pH was adjusted using 1 M sodium hydroxide (Th. Geyer GmbH Co. KG, Renningen, Germany). Sodium carbonate was acquired from Merck KGaA, Darmstadt, Germany.

An aqueous PS nanoparticle size standard with a nominal diameter of 100 nm ± 3 nm (TEM) at a concentration of 1% (*w*/*w*) (Nanosphere™ Size Standard 3100A, Thermo Fisher Scientific, Waltham, MA, USA) was used.

A commercially available liposome sample, liposomal Doxorubicin HCL (Caelyx pegylated liposomal formulation) was obtained from Johnson & Johnson Romania SRL, Bucharest, Romania. The liposome concentration within the formulation was 2 mg mL^−1^. Furthermore, a lyophilized exosome standard from HansaBioMed Life Science (HBM-PEU-100/2) Tallinn, Estonia, where exosomes were extracted from human urine, was used for the exosome experiments. The nominal concentration was 8.2 × 10^11^ particles mL^−1^.

A DMEM was purchased from Invitrogen, Germany and prepared with the subsequent components, 10% FCS (Invitrogen, Germany), 1% non-essential amino acids (Invitrogen, Karlsruhe, Germany) and 1% penicillin/streptomycin (Invitrogen, Karlsruhe, Germany). Throughout this manuscript, the abbreviation DMEM was used to describe the complete medium including 10% FCS. Additionally, rabbit serum was acquired from Sigma Aldrich, Taufkirchen, Germany.

### 3.2. Sample Preparation

The PS size standard was gravimetrically diluted in UPW to obtain a concentration of 2.5 × 10^8^ particles mL^−1^ for the qualification measurements of the NTA system. For the EAF4-MALS-NTA measurements, a concentration of around 20 mg L^−1^ of PS beads in UPW was prepared. The experiments regarding the DMEM cell culture medium were conducted with the same PS concentration of around 20 mg L^−1^ using DMEM instead of UPW for dilution.

For the liposomal Doxorubicin HCl EAF4-MALS-NTA experiments, a suspension with a final concentration of around 0.67 mg L^−1^ in the respective carrier solution was produced. The liposome sample in DMEM cell culture medium was prepared accordingly by diluting directly in DMEM cell culture medium in place of the carrier solution.

The exosome pellet was recovered according to the manufacturer’s guidelines, using only UPW. All exosome experiments were carried out with a final exosome concentration of 1.64 × 10^10^ particles mL^−1^. For this purpose, the samples were diluted in UPW. In order to obtain the exosome rabbit serum samples, the dilution of the exosome standard solution was carried out directly in rabbit serum with a 1:10 dilution in UPW prior to injection obtaining a concentration of 1.64 × 10^10^ particles mL^−1^.

### 3.3. Instrumentation

After preparation, the samples were fractionated by an EAF4 system from Postnova Analytics (EAF2000 MT, Postnova Analytics GmbH (PN), Landsberg am Lech, Germany) including an autosampler (PN 5300), Slot Outlet (PN1650) and channel thermostat (PN 4020), which was equipped with an electrical analytical fractionation channel. The applied electrical field was controlled by an Electrical FFF Module (PN2410). A regenerated cellulose membrane of 10 kDa molecular weight cut-off and a Mylar spacer of 350 µm height were placed in the separation channel, which had a tip-to-tip length of 277 mm. The temperature of the channel thermostat was set to 25 °C, whereas the samples were kept at 6 °C in the autosampler. Fractions were collected using a directly coupled fraction collector (PN8050). The EAF4 system was hyphenated with a UV-vis detector (PN3211, Postnova Analytics GmbH, Landsberg am Lech, Germany) and a MALS detector (PN3621, 21 angles, Postnova Analytics GmbH, Landsberg am Lech, Germany) whose scattering angles were normalized with respect to the 92° angle measuring a 61 nm PS-size standard fractionated by EAF4. The instrument control and data analysis of the EAF4, as well as the data evaluation of the MALS detector, were performed by the NovaFFF AF2000 Control software (Version 2.1.0.4, Postnova Analytics, Landsberg, Germany). 19 active angles from 12° to 156° were evaluated for the results. The scattering data from the PS size standard as well as the liposome and exosome samples (17 angles) were fitted to a spherical model. A fraction collector (PN8050, Postnova Analytics GmbH, Landsberg am Lech, Germany) was used to collect defined size fractions after separation. The fractions were analyzed by NTA.

For all matrices, sample recoveries were determined by comparing the areas under the respective peaks obtained from PS100 beads and liposomal Doxorubicin HCl both in presence and absence of the crossflow/electrical field during EAF4 separation using a UV-vis detector at a wavelength of 254 nm (PS100) respectively 362 nm (liposomal Doxorubicin HCl) (data not shown). The increased recovery of the PS100-DMEM samples may be caused by the absorbed proteins forming the protein corona, which may contribute to the UV-vis absorbance at 254 nm wavelength (Table 2). Due to a low UV-vis signal of the exosomes in both matrices the determination of the recovery of the exosome samples was not feasible and therefore not determined.

For the determination of the electrophoretic mobility, a series of EAF4 measurements with varying electrical field strengths (between ± 10 V m^−1^) were applied, while keeping the other fractionation parameters constant (Appendix A). The electrophoretic mobility was calculated by plotting the drift velocity that resulted from the observed shift in retention time compared to a reference measurement without any electrical field, against the applied field strength (Appendix A). The applied field strength depends on the applied constant current and on the conductivity of the carrier solution. Both parameters are evaluated and monitored over the complete fractionation run. After a linear least squares regression analysis of the plotted data, the electrophoretic mobility is obtained from the slope of the linear regression line. The Zeta potential can be determined from the electrophoretic mobility using the Smoluchowski theory and Smoluchowski approximation. The evaluation was carried out in the NovaAnalysis software (Version 2007, Postnova Analytics GmbH, Landsberg, Germany).

A NanoSight NS300 (NTA) system equipped with a laser with a wavelength at 405 nm and a high sensitivity scientific complementary metal-oxide-semiconductor (sCMOS) camera from Malvern Panalytical Instruments Ltd. (Malvern, UK) was used.

The NanoSight syringe pump was only used for system qualification and offline experiments. NanoSight LVFC was installed after a flow splitter to allow a connection with the MALS flow cell. The system performance was checked, and the focus was determined offline at the beginning of each day by measuring 100 nm PS beads, thereby a qualification was considered adequate with size deviations below 6% (as advised by ISO 19430 standard) [39]. The camera level was evaluated for each sample type individually using the automatic software settings by measuring it offline to take different optical properties into account. For the offline experiments, a syringe pump with syringe speed of 50 a.u. was used to transfer the sample into the measurement cell (LVFC) and five times 60 s videos were captured.

In comparison to that, a capturing time of 30 s was used for online measurements. To start the NTA scripts and therefore the captures, the MALS signal was used to detect the relevant signals. As a consequence, the total collection time for one data point consisted of 30 s plus around 1 s for restarting a new capture. Each video corresponded to the data of a 31 s time interval. Additionally, the signals were corrected for the time delay caused by the tubing connecting the individual detectors. The flow principle is illustrated in Figure 7. Advanced scripting was used for automated captures and processing of all samples after each EAF4 measurement. The detection threshold for video processing was kept constant for all experiments at 5. All used NTA scripts are presented in the Appendix A. For instrument control, data evaluation and calculation, the software version 3.4.003 (Malvern Panalytical Instruments Ltd., Malvern, UK) was used. A summary file containing all information obtained from all videos was created and used for further data processing.

Further data processing (averaging, plotting etc.) was performed using OriginPro 2018b (OriginLab Corporation, Northampton, MA, USA).

To obtain a sufficiently low flow rate in NTA flow cell, the channel flow rate was reduced from 0.50 mL min^−1^ to a detector flow rate of 0.15 mL min^−1^ by using the slot outlet technology (see Figure 7 and Appendix A). The slot outlet technology takes advantage of the fact that during separation, analytes and sample constituents are located within a region very close to the accumulation wall [36,37]. As a consequence, the band above this sample region includes no analytes and this carrier stream can be removed without losing sample constituents. Next to reducing the detector flow, another advantage of this technique is the concentrating of sample after dilution within the separation channel. After passing the MALS detector, the flow rate was further reduced by a flow splitter to around 12 µL min^−1^ on average, which was determined gravimetrically. Furthermore, the traveling time of particles through the analysis window of the NTA was between 5 to 10 s, which is recommended. The flow splitter consists of a T-piece, where the inlet tubing is connected to the outlet of the MALS detector. The splitting ratio was adjusted by the ratio of the length of both outlet tubing, both with an ID of 100 µm. This is a so-called static flow splitter. For minimizing band broadening caused by inter-detector backmixing effects, the tubing lengths were optimized to a minimum length and the LVFC was connected at the “short” end, see Appendix A. No meaningful flow rate variations were observed using this static flow splitter over the applied pressure range during EAF4 experiments. The pressure inside the channel was adjusted to 3 to 4 bars with only small changes during a fractionation experiment. Especially during the time period of NTA video capturing the system pressure was not changing significantly resulting in constant flow rates inside the LVFC. In conclusion, the two main parts, both static flow splitter and slot outlet technique, resulted in an effective and robust combination to sufficiently lower the LVFC flow rate for all three fractionation conditions, while maintaining an adequate flow rate for MALS detection.

The potential effect of band broadening, which may occur during EAF4 separation, was not further investigated as researchers previously proved this effect to be negligible when using an advanced AF4 focusing step prior to size determination by MALS [44].

The method validation was performed after fractionation and collection of a defined fraction of PS samples.

### 3.4. Fractionation Method

The EAF4 injection and focusing step were initiated with a 7 min-long injection flow rate of 0.15 mL min^−1^ and an initial cross flow rate of 1.00 mL min^−1^. After a 0.2 min-long transition time, the elution step was carried out with a constant cross flow rate of 1.00 mL min^−1^ for 1 min, followed by an power decay (exponent 0.2) for 40 min to 0.10 mL min^−1^. The elution ended with a 2 min long constant cross flow rate at 0.10 mL min^−1^. A 10 min long rinse step was carried out after the elution. The channel flow rate was constant at 0.50 mL min^−1^. The described fractionation method was used to separate the PS size standard. A 0.4 mM sodium carbonate solution was employed as a carrier solution. All fractionation programs are summarized in Appendix A.

Two optimized elution profiles for the liposomes (“Fractionation method B”) and exosomes (“Fractionation method C”) were developed. Firstly, the focusing step for the liposomes consisted of a 0.08 mL min^−1^ injection flow rate for 7 min with an initial cross flow rate of 0.7 mL min^−1^ and a transition time of 12 s after the injection. A channel flow rate of 0.5 mL min^−1^ with a linear cross flow decay over 30 min was used for the fractionation followed by a 10 min-long rinse step without cross flow. The carrier solution consisted of a 0.5 mM sodium chloride solution.

For separating the exosomes, the elution profile was optimized for a separation of a broader size distribution. A channel flow rate of 0.50 mL min^−1^ was applied; the injection and focusing step were performed with an injection flow rate of 0.2 mL min^−1^, an initial cross flow rate of 1.00 mL min^−1^ and an injection and focusing time of 6 min. After a 30 s long transition time the elution began with a 25 min linear cross flow decay reaching 0.28 mL min^−1^ cross flow rate, followed by two 5 min long power decays with an exponent of 0.8 to a cross flow rate of 0.15 mL min^−1^, respectively 0.08 mL min^−1^. The final elution step was reached after a 5 min long power decay (exponent 0.9) to 0.05 mL min^−1^ and was kept constant for 30 min. Afterwards a rinse step of 20 min was used to ensure reproducible results with flushing the system and removing potential higher aggregates. The exosome experiments were performed with a carrier solution of a PBS buffer at a pH of 7.4.

## 4. Conclusions

In the present work, we describe the successful hyphenation of Electrical Asymmetrical Flow Field-Flow Fractionation coupled to Multi-Angle Light Scattering detection with Nanoparticle Tracking Analysis (EAF4-MALS-NTA). Hyphenation was realized taking advantage of the slot outlet option of the EAF4 channel and a simple T-piece used as a flow splitter between the MALS detector and the NTA in order to ensure a compatible flow rate between the two systems. While EAF4-MALS enabled access to radius of gyration and electrophoretic mobility (Zeta potential), NTA provided valuable additional information about hydrodynamic diameter and particle number concentration of three different, nano-sized samples investigated in matrices of various complexity.

After a successful validation using well-defined polystyrene nanoparticles, the performance of the EAF4-MALS-NTA setup was tested using liposomes and exosomes spiked into cell culture medium and rabbit serum, respectively. Obtained results clearly demonstrate the benefit of an upstream EAF4 separation and purification step to separate and remove potentially interfering matrix components from the liposome and exosome sample prior to NTA analysis while additionally providing valuable insight into their surface charge and thus potential interactions with matrix components in these complex biological media.

The presented EAF4-MALS-NTA setup is a powerful analytical platform for the comprehensive physico-chemical characterization of nano-objects of various compositions in complex media with both techniques mutually benefiting from each other. While NTA represents a true particle counting detector for EAF4-MALS also enabling access to particle shape analysis, the online sample purification and matrix removal capabilities of EAF4 help to overcome the limitations that NTA faces when applied to sample matrices with a high light scattering background. The here described setup therefore is a promising analytical tool to predict and study particle behavior and interactions with media components even under in vivo like conditions.

## Figures and Tables

**Figure 1 molecules-25-04703-f001:**
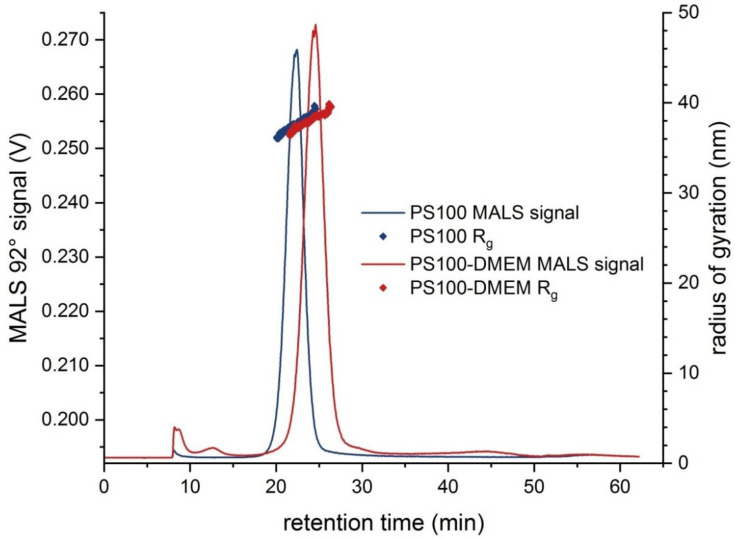
EAF4-MALS fractograms for PS100 sample (**blue**) in UPW, PS100-DMEM sample (**red**) respectively. The MALS 92° signal (**line**) is overlaid by R_g_ (**dots**) distribution (electrical field strength 0.0 V m^−1^).

**Figure 2 molecules-25-04703-f002:**
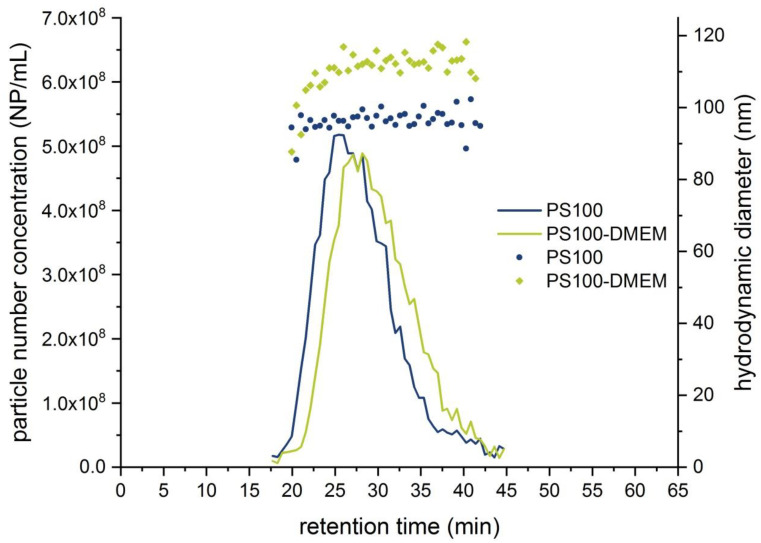
EAF4-NTA fractograms displaying the particle number concentrations (**lines**), respectively D_h_ (**dots**), for the sample PS100 (**blue**) and PS100-DMEM (**green**) (V(inj) = 25 µL, 0.0 V m^−1^).

**Figure 3 molecules-25-04703-f003:**
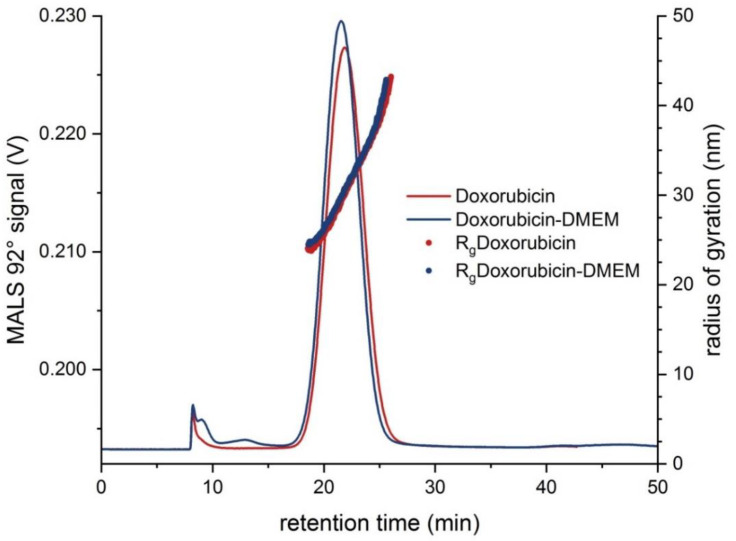
EAF4-MALS fractograms for liposomal Doxorubicin HCl (**red line**) and Doxorubicin-DMEM (**blue line**) overlaid with R_g_ (**red** and **blue dots**, respectively) at a mass concentration of 13 mg L^−1^ (0.0 V m^−1^, *n* = 3).

**Figure 4 molecules-25-04703-f004:**
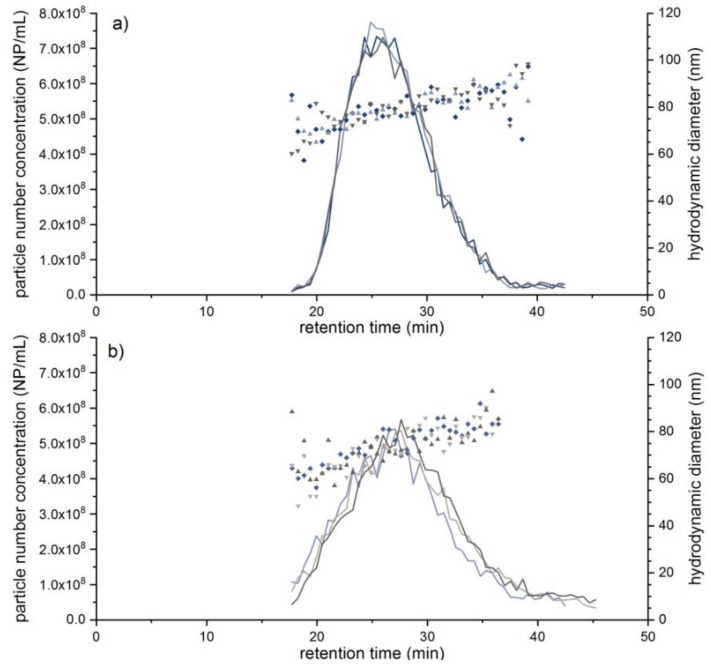
(**a**) EAF4-NTA fractograms of liposomal Doxorubicin (c = 0.67 mg L^−1^, V(inj) = 40 µL, 0.0 V m^−1^, *n* = 3) overlaying obtained particle number concentrations (**lines**) and D_h_ (**dot plot**). (**b**) EAF4-NTA fractograms of liposomal Doxorubicin diluted in DMEM medium (c = 0.42 mg L^−1^, V(inj) = 20 µL, 0.0 V m^−1^, *n* = 3) overlaying obtained particle number concentrations (**lines**) and D_h_ (**dot plot**).

**Figure 5 molecules-25-04703-f005:**
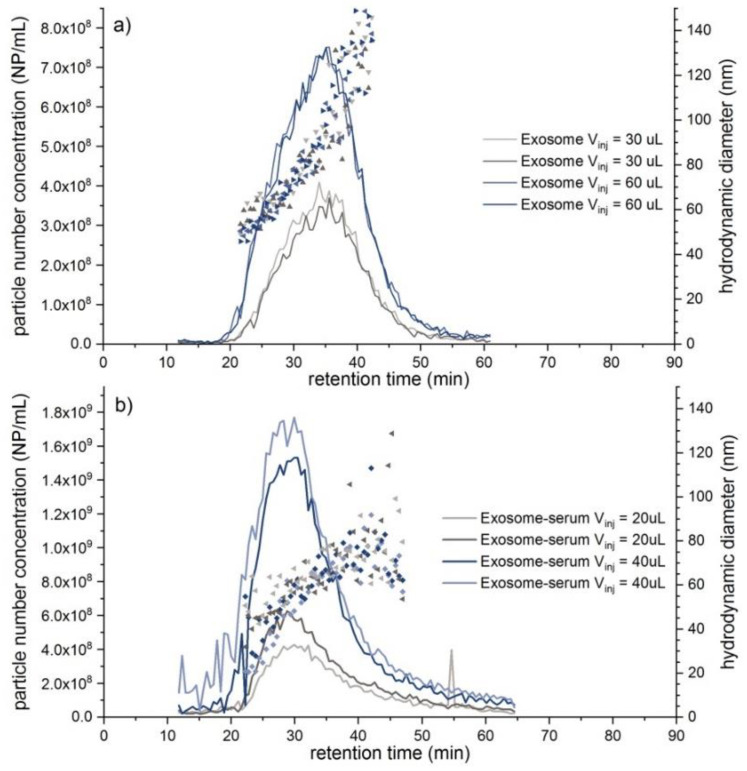
(**a**) EAF4-NTA fractograms of an exosome standard (V(inj) = 30 µL, respectively 60 µL, 0.0 V m^−1^) overlaying obtained particle number concentrations (**lines**) and D_h_ (**dot plot**). (**b**) EAF4-NTA fractograms of the exosome-serum sample (V(inj) = 20 µL, respectively 40 µL, 0.0 V m^−1^) overlaying particle number concentration (**lines**) with D_h_ distribution (**dot plot**).

**Figure 6 molecules-25-04703-f006:**
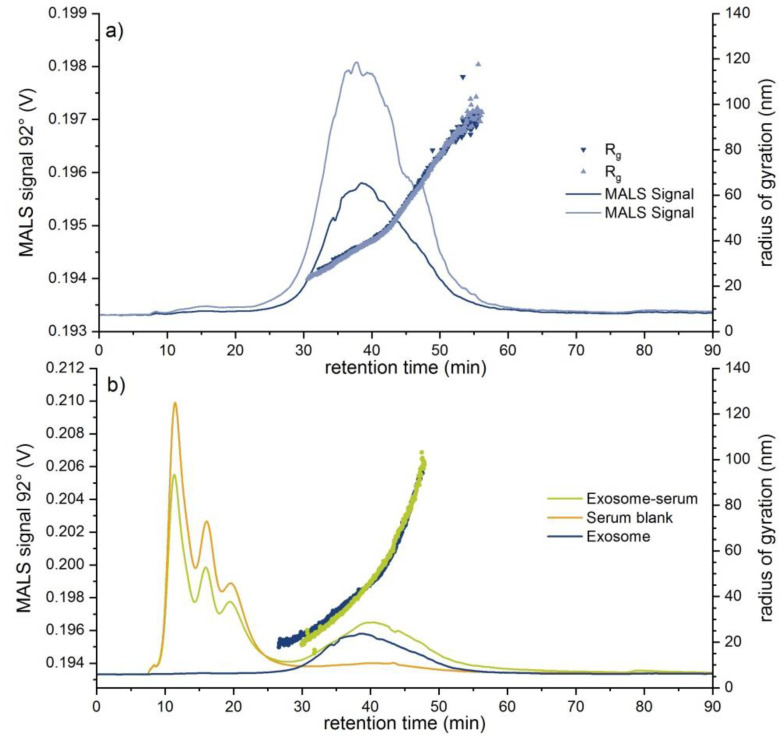
(**a**) EAF4-MALS fractograms of an exosome standard sample (V(inj) = 30 µL, respectively 60 µL, 0.0 V m^−1^) overlaying MALS 92° signal (**lines**) with R_g_ distribution (**dot plot**). (**b**) Overlay of different measurements (V(inj) = 30 µL) comparing MALS signals and R_g_’s (respective dots) from the exosome-serum sample (**green curve**) with the exosome standard (**blue curve**) and the serum blank (**brown**).

**Figure 7 molecules-25-04703-f007:**
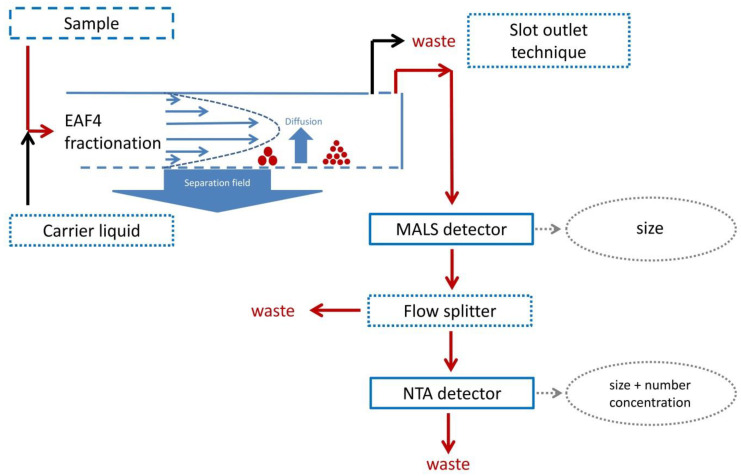
Schematic principle of the EAF4-MALS-NTA setup.

**Table 1 molecules-25-04703-t001:** Results of offline NTA analysis of fractions collected after EAF4 separation.

Sample	Injected Sample Volume V(inj)	Hydrodynamic Diameter D_h_		Measured Particle Number Concentration	
	(µL)	Mode (nm)	±Relative (%)	(NP mL^−1^)	±Relative (%)
PS100	25	97.5	0.6	3.75 × 10^8^	1.62
	10	99.3	0.3	1.53 × 10^8^	4.20
PS100-DMEM	25	117.7	0.8	3.28 × 10^8^	1.73
	10	118.1	0.6	1.56 × 10^8^	3.93

**Table 2 molecules-25-04703-t002:** Obtained EAF4-MALS-NTA results for the PS100 and PS100-DMEM samples.

Sample	MALS Results		NTA Results			EAF4 Evaluation	
	R_g_ at Peak Maximum (nm)	R_g_ Range (nm)	D_h_ at Peak Maximum (nm)	D_h_ Range (nm)	Particle Concentration at Peak Maximum (NP mL^−1^)	Zeta Potential (mV)	Recovery (%)
PS100	37.5 ± 0.2	36–40	96.7 ± 0.5	95–102	5.20 × 10^8^	−48.0 ± 3.0	90.7 ± 1.4
PS100-DMEM	37.3 ± 0.5	36–40	113.3 ± 3.0	105–119	4.90 × 10^8^	−37.1 ± 1.9	100.0 ± 1.0

**Table 3 molecules-25-04703-t003:** EAF4-MALS-NTA results for liposomal Doxorubicin HCl and Doxorubicin-DMEM samples.

Sample	MALS Results		NTA Results			EAF4 Evaluation	
	R_g_ at Peak Maximum (nm)	R_g_ Range (nm)	D_h_ at Peak Maximum (nm)	D_h_ Range (nm)	Particle Concentration at Peak Maximum (NP mL^−1^)	Zeta Potential (mV)	Recovery (%)
Liposomal Doxorubicin HCl	30.4 ± 0.7	24–41	76.9 ± 1.9	65–95	7.50 × 10^8^	−34.6 ± 1.5	87.6 ± 1.6
Doxorubicin-DMEM	29.9 ± 0.3	24–41	72.9 ± 3.0	60–95	5.40 × 10^8^	−45.2 ± 1.5	89.9 ± 1.9

**Table 4 molecules-25-04703-t004:** EAF4-MALS-NTA results for the exosome standard and the exosome-serum samples. The Zeta potential could not be assessed for either sample.

Sample	MALS Results		NTA Results			EAF4 Evaluation	
	R_g_ at Peak Maximum (nm)	R_g_ Range (nm)	D_h_ at Peak Maximum (nm)	D_h_ Range (nm)	Particle Concentration at Peak Maximum (NP mL^−1^)	Zeta Potential (mV)	Recovery (%)
Exosome Standard	37.9 ± 1.5	23–100	98.2 ± 6.8	43–150	7.50 × 10^8^	n.d.	n.d.
Exosome-serum	43.3 ± 1.0	23–100	41.2 ± 11.9	35–90	1.60 × 10^9^	n.d.	n.d.

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
