# Peer review of "Fast and Purification-Free Characterization of Bio-Nanoparticles in Biological Media by Electrical Asymmetrical Flow Field-Flow Fractionation Hyphenated with Multi-Angle Light Scattering and Nanoparticle Tracking Analysis Detection"

_molecules, 2020, doi:10.3390/molecules25204703_

Round 1
Reviewer 1 Report
The work performed by Drexel and al. describe a unique hyphenated approach combining fractionaction of particles by size and surface charge by EAF4, sizing (NTA, MALS) and online particle concentration measurements (NTA). The advanced approach presented by the authors allows to combine particle separation, sizing measurement of surface charge and concentration in simple and complex media, being a very versatile tool for characterisation of nanoparticles physico-chemical properties and particle stability in complex biological media. It may be of great interest in the field of nanomedicine for the development (screening) of new formulations, for stability studies and for QC purposes. I would suggest to consider the manuscript for publication with some minor revisions. Prior to publication, the manuscript will benefit from a moderate English language and style revision.
Additional minor revisions suggested:
- shortening the title
- Table 1 reporting the error on the measurement in the same format for sizing and concentration (e.g. either SD or CV%)
- Figures & Tables: suggest reporting all the sizing values either in radius or in diameter for a more immediate and direct comparability (instead of NTA in diameter and MALS in radius). The same applied to the values reported in the result and discussion.
- It would be useful to have the conclusion and perspective section following the results and discussion (instead of having R&D, M&M and then conclustions).
Other minor corrections suggested:
- Line 37: pre-formulation screening, could be rephrased as in formulation preliminary screening, or simply with formulation optimization?
- Line 39: " in the lead candidate identification for a later evaluation in more complex formulations" I would consider rephasing this sentence
- Line 50: "Similarly, characterization of bio-particles like exosomes or viruses extracted from various body fluids brings the need of validation of purification and isolation techniques." Please rephrase. Each method in the pharmaceutical settings require validation. Characterisation of exosomes and virus require an isolation and purification step, as an additional step.
- Line 58 "While supporting biological assays, analytical techniques give only partial information of a sample under specific conditions, and in many cases, “clean up” processes are required to enable best measurement conditions for the technology". Do you mean that in complex media, additional complexity is caused by the need of particle isolation from the media prior to analytical analysis. If so, please consider rephrasing
- Line 60: Multi-technology characterization? Do you mean the combination of complementary techniques (separate measurements) or hyphenated techniques or both? Please clarify
- Line 77: the possibility to remove matrix components smaller than the molecular weight cutoff of the ultrafiltration membrane via the cross flow renders EAF4 (just like AF4) a promising tool for an online purification of complex biological samples prior to further multi-detector analysis. Not clear, please reformulate: with 10 kDa membranes, serum proteins are not filtered (can not pass through), are fractionated and exit in the void before particles, so technically are not removed are separated from the particle population. Other media components 8antibiotics, sugar etc could pass through the semipermeable membrane
- Line 112: I would suggest to briefly explain the rational for using polystyrene as reference particles
- Line 118: Dulbecco’s Modified Eagle’s 118 Medium (DMEM) completed with serum (so complete media) or without serum proteins? Please specify if you are referring to complete media. Same in line 148, and in the rest of the manuscript.
- Line 208: It was observed that the presence of proteins in the DMEM cell culture medium did not significantly affect the Dh of the liposomes indicating no measurable formation of a protein corona on the liposomes’ surface at least for the duration of the whole experiment (< 2 h). The absence of size shift in presence of plasma proteins (no significant protein interactions) in the case of liposomal doxorubicin are in agreement with the data published in the following papers: doi: 10.1007/s00216-019-02252-9, doi: 10.2217/nnm-2017-0338. Mentioning this could be interesting for the discussion. The novelty of this work (that can be stressed by the author) is the capability to combine online sizing measurements with zeta potential and particle concentration analysis, which give more insight into understanding the particle interactions with media components.
- Line 310: was gravimetrically diluted in UPW to around5E+8 particles mL. If preparation was made gravimetrically, and the concentration values could be derived precisely I would suggest to eliminate "around" from the sentence. Otherwise is to around 3E+08
- Line 344: please specify the wavelength used for recovery calculations
Author Response
First of all, we would like to express our sincere gratitude to the editor and all three reviewers for their time and valuable comments and suggestions to improve the quality of the manuscript. We have addressed all of them as comprehensive as possible and revised the manuscript accordingly. Our respective answers highlighted in bold can be found below. For a better tracing, we also highlighted our changes in the manuscript using the Word “track changes” function.
Answers to the comments from the reviewers
Reviewer 1
The work performed by Drexel and al. describe a unique hyphenated approach combining fractionaction of particles by size and surface charge by EAF4, sizing (NTA, MALS) and online particle concentration measurements (NTA). The advanced approach presented by the authors allows to combine particle separation, sizing measurement of surface charge and concentration in simple and complex media, being a very versatile tool for characterisation of nanoparticles physico-chemical properties and particle stability in complex biological media. It may be of great interest in the field of nanomedicine for the development (screening) of new formulations, for stability studies and for QC purposes. I would suggest to consider the manuscript for publication with some minor revisions. Prior to publication, the manuscript will benefit from a moderate English language and style revision.
Additional minor revisions suggested:
Shortening the title
The authors also had the feeling that the title might be a bit too bulky. So we suggest the following alternative:
“Fast and purification-free characterization of bio-nanoparticles in biological media by Electrical Asymmetrical Flow Field-Flow Fractionation hyphenated with Multi-Angle Light Scattering and Nanoparticle Tracking Analysis”
We agree that this is not much of a reduction in length, but this is mainly due to the fairly long description of the technologies used in this study, which we don’t want to abbreviate as these represent the essential core of this study.
Table 1 reporting the error on the measurement in the same format for sizing and concentration (e.g. either SD or CV%)
The error of the size was changed to relative values to be in accordance with the error of the concentration measurements.
Figures & Tables: suggest reporting all the sizing values either in radius or in diameter for a more immediate and direct comparability (instead of NTA in diameter and MALS in radius). The same applied to the values reported in the result and discussion.
We agree that it might be a bit confusing to use hydrodynamic diameter and radius of gyration in parallel; however these are the analytical endpoints that are usually displayed when MALS and/or NTA are used in the respective scientific literature. So we prefer to use these terms in order to provide an easier comparability with other studies.
Furthermore, even though both parameters describe the size of a respective object, radius of gyration and hydrodynamic size are actually not directly comparable (or at least not without consideration of defined shapes) due to differences in the underlying fundamental physical principles.
We do, however, use the term diameter of gyration (Dg = 2 x Rg) later in the manuscript (e.g. line 228) for the calculation of the shape factor (shape factor ρ = Dg/Dh).
It would be useful to have the conclusion and perspective section following the results and discussion (instead of having R&D, M&M and then conclusions).
We also agree with the reviewer’s opinion here, however, the order of the sections was defined by the journal and could not be changed.
Other minor corrections suggested:
Line 37: pre-formulation screening, could be rephrased as in formulation preliminary screening, or simply with formulation optimization?
Thank you, we have rephrased this sentence accordingly (see our answer below).
Line 39: " in the lead candidate identification for a later evaluation in more complex formulations" I would consider rephrasing this sentence
This sentence is now rephrased to:
“One of the key stages in therapeutics’ development is formulation optimization to select the right candidate, where analytical techniques support lead candidate identification for later evaluation in more complex formulations.”
Line 50: "Similarly, characterization of bio-particles like exosomes or viruses extracted from various body fluids brings the need of validation of purification and isolation techniques." Please rephrase. Each method in the pharmaceutical settings require validation. Characterisation of exosomes and virus require an isolation and purification step, as an additional step.
Thanks for this suggestion! We have rephrased the sentence accordingly:
“Similarly, when characterizing bio-particles extracted from various body fluids, e.g. exosomes or viruses, additional isolation and purification steps are necessary, which also require additional validation steps.”
Line 58 "While supporting biological assays, analytical techniques give only partial information of a sample under specific conditions, and in many cases, “clean up” processes are required to enable best measurement conditions for the technology". Do you mean that in complex media, additional complexity is caused by the need of particle isolation from the media prior to analytical analysis. If so, please consider rephrasing
Thanks for this suggestion! We have rephrased the sentence accordingly:
“While supporting biological assays, analytical techniques provide information on a sample under test specific conditions. In many cases for biological nanoparticles the samples are a complex mixture, so isolation and purification processes are required to enable best measurement conditions for the technology, to allow high quality decision-making data to be generated”
Line 60: Multi-technology characterization? Do you mean the combination of complementary techniques (separate measurements) or hyphenated techniques or both? Please clarify
We have added a respective section to this sentence for clarification:
“Multi-technology characterization, meaning the combination of complementary analytical and bioassay techniques, aids in expanding knowledge…”
Line 77: the possibility to remove matrix components smaller than the molecular weight cutoff of the ultrafiltration membrane via the cross flow renders EAF4 (just like AF4) a promising tool for an online purification of complex biological samples prior to further multi-detector analysis. Not clear, please reformulate: with 10 kDa membranes, serum proteins are not filtered (cannot pass through), are fractionated and exit in the void before particles, so technically are not removed are separated from the particle population. Other media components 8antibiotics, sugar etc could pass through the semipermeable membrane
This is correct, we have explained this in more detail to emphasize on both, removal of small components and separation of matrix components from the analyte. The sentence now reads as follows:
“Besides offering high-resolution separation and access to particle size distribution and surface charge (Zeta potential) of a respective sample, the possibility to purify (i.e. to remove matrix components smaller than the molecular weight cutoff of the ultrafiltration membrane via the cross flow) and separate matrix components from the analyte of interest renders EAF4 (just like AF4) a promising tool for an online purification of complex biological samples prior to further multi-detector analysis.”
Line 112: I would suggest to briefly explain the rational for using polystyrene as reference particles
Polystyrene nanoparticles are the ideal model particle system to better understand and evaluate the performance and this novel hyphenation as they are:
- Already well-characterized by various analytical techniques (certified by NIST)
- Are used to normalize the angular dependent MALS intensities
- Are used to check status and performance of the NTA system
Moreover, it is well-known that polystyrene nanoparticles form a protein corona in complex biological matrices, as pointed out by Contado et al. (2019) for example (see Ref 38). They are therefore the ideal candidate to highlight the unique capabilities of the presented methodology.
We have rephrased the first sentence of this paragraph in order to explain the rationale behind this choice as follows:
“In a first step the hyphenation of EAF4-MALS and NTA was validated against well-characterized polystyrene nanoparticles, which served as a model particle system to evaluate the suitability of this novel setup for the analysis of nanoparticles in complex biological media.”
Line 118: Dulbecco’s Modified Eagle’s 118 Medium (DMEM) completed with serum (so complete media) or without serum proteins? Please specify if you are referring to complete media. Same in line 148, and in the rest of the manuscript.
In all experiments a full medium containing the cell culture medium together with 10 % FCS was used, see also section 3.1. We have clarified this with the first naming of DMEM and we have rephrased the respective sentence in 3.1.
Line 208: It was observed that the presence of proteins in the DMEM cell culture medium did not significantly affect the Dh of the liposomes indicating no measurable formation of a protein corona on the liposomes’ surface at least for the duration of the whole experiment (< 2 h). The absence of size shift in presence of plasma proteins (no significant protein interactions) in the case of liposomal doxorubicin are in agreement with the data published in the following papers: doi: 10.1007/s00216-019-02252-9, doi: 10.2217/nnm-2017-0338. Mentioning this could be interesting for the discussion. The novelty of this work (that can be stressed by the author) is the capability to combine online sizing measurements with zeta potential and particle concentration analysis, which give more insight into understanding the particle interactions with media components.
Thank you for the references. We have added a sentence to point to the references to support our observations (line 215f.).
Line 310: was gravimetrically diluted in UPW to around5E+8 particles mL. If preparation was made gravimetrically, and the concentration values could be derived precisely I would suggest to eliminate "around" from the sentence. Otherwise is to around 3E+08
This is correct and more precise, we have removed “around”.
Line 344: please specify the wavelength used for recovery calculations
The information about the wavelength was added to the respective paragraph. The evaluation of the recovery for the PS100 and PS100-DMEM samples was carried out using a wavelength of 254 nm. Whereas, a wavelength of 362 nm was applied during the liposome measurements.
Reviewer 2 Report
This paper is very well written and presents a subject, which can be of high interest in de health areas. Im my point of view, it may be published without modifications.
Author Response
First of all, we would like to express our sincere gratitude to the editor and all three reviewers for their time and valuable comments and suggestions to improve the quality of the manuscript. We have addressed all of them as comprehensive as possible and revised the manuscript accordingly. Our respective answers highlighted in bold can be found below. For a better tracing, we also highlighted our changes in the manuscript using the Word “track changes” function.
Answers to the comments from the reviewers
Reviewer 2
The authors are grateful for this very positive feedback and would like to thank the reviewer for his/her time and effort.
Reviewer 3 Report
Nice, well written paper. Interesting, and highly relevant.
A few very minor editorial comments:
Pg.1, line 26 “coupling of the two systems”, rather than “coupling of both systems”.
Pg. 2, line 44 ff: “Therefore, characterization of bioparticles and drug carrier systems in conditions mimicking the in vivo environment is gaining an interest, along with an earlier understanding of their toxicological effects.“
The meaning of this is unclear. Is the intention to highlight the interest in determining the toxicological effects as early as possible or is this referring to prior knowledge about their toxicological effects?
Pg 2, line 49: Consistency in the word choice: bioparticles or bio-particles?
Pg 12, line 367: “…deviations below 6 % (as advised by ISO 19430 standard [39]. “ <=missing bracket
Pg 18, line 512 and line 525: there is an error in the script description, assuming Script C (maybe also in bold?) was used for the exosomes.
“Script B was used for the samples liposomal Doxorubicin HCl and Doxorubicin-DMEM: “
“Script C was used for the liposomal Doxorubicin HCl and Doxorubicin-DMEM: “
Pg 24, 581: There is no reference to nor explanation for Fig S9….
Author Response
First of all, we would like to express our sincere gratitude to the editor and all three reviewers for their time and valuable comments and suggestions to improve the quality of the manuscript. We have addressed all of them as comprehensive as possible and revised the manuscript accordingly. Our respective answers highlighted in bold can be found below. For a better tracing, we also highlighted our changes in the manuscript using the Word “track changes” function.
Answers to the comments from the reviewers
Reviewer 3
Nice, well written paper. Interesting, and highly relevant.
A few very minor editorial comments:
Pg.1, line 26 “coupling of the two systems”, rather than “coupling of both systems”.
This part was changed.
Pg. 2, line 44 ff: “Therefore, characterization of bioparticles and drug carrier systems in conditions mimicking the in vivo environment is gaining an interest, along with an earlier understanding of their toxicological effects.“
The meaning of this is unclear. Is the intention to highlight the interest in determining the toxicological effects as early as possible or is this referring to prior knowledge about their toxicological effects?
The intention was to highlight that both aspects are important. We have therefore revised this sentence as follows:
“Therefore, characterization of bio-particles and drug carrier systems in conditions mimicking the in vivo environment is gaining an interest. In addition, easily and quickly obtaining an earlier understanding of their toxicological effects is attractive within the therapeutic development pipeline.”
Pg 2, line 49: Consistency in the word choice: bioparticles or bio-particles?
The term bio-particles is now used throughout the complete manuscript.
Pg 12, line 367: “…deviations below 6 % (as advised by ISO 19430 standard [39]. “ <=missing bracket
We have added the missing bracket.
Pg 18, line 512 and line 525: there is an error in the script description, assuming Script C (maybe also in bold?) was used for the exosomes.
“Script B was used for the samples liposomal Doxorubicin HCl and Doxorubicin-DMEM: “
“Script C was used for the liposomal Doxorubicin HCl and Doxorubicin-DMEM: “
There is a mistake, we have corrected it. “Script C” was used to analyze the exosome and exosome-serum samples.
Pg 24, 581: There is no reference to nor explanation for Fig S9….
The Figures are now referenced in the respective section (Section 3.3). We have changed the order of Figures S5-S8 and Figures S9-S12.
Figure S5, S6, S7, S8 show fractograms with different electrical field strengths and induced retention time shifts. We have added a more detailed description to the figures, which explains the differences and the legend of each fractogram. The retention time shift was used to calculate the drift velocity that was used to calculate the electrophoretic mobility, respectively zeta potential.